# Targeting Glucose Metabolism: A Novel Therapeutic Approach for Parkinson’s Disease

**DOI:** 10.3390/cells13221876

**Published:** 2024-11-13

**Authors:** Ahmed Tanvir, Junghyun Jo, Sang Myun Park

**Affiliations:** 1Department of Pharmacology, Ajou University School of Medicine, Suwon 16499, Republic of Korea; tanvir203@ajou.ac.kr (A.T.); junghyunjo@ajou.ac.kr (J.J.); 2Neuroscience Graduate Program, Department of Biomedical Sciences, Ajou University School of Medicine, Suwon 16499, Republic of Korea; 3Center for Convergence Research of Neurological Disorders, Ajou University School of Medicine, Suwon 16499, Republic of Korea

**Keywords:** neurodegenerative disease, Parkinson’s disease, glucose metabolism, anti-diabetic drugs, drug repositioning

## Abstract

Glucose metabolism is essential for the maintenance and function of the central nervous system. Although the brain constitutes only 2% of the body weight, it consumes approximately 20% of the body’s total energy, predominantly derived from glucose. This high energy demand of the brain underscores its reliance on glucose to fuel various functions, including neuronal activity, synaptic transmission, and the maintenance of ion gradients necessary for nerve impulse transmission. Increasing evidence shows that many neurodegenerative diseases, including Parkinson’s disease (PD), are associated with abnormalities in glucose metabolism. PD is characterized by the progressive loss of dopaminergic neurons in the substantia nigra, accompanied by the accumulation of α-synuclein protein aggregates. These pathological features are exacerbated by mitochondrial dysfunction, oxidative stress, and neuroinflammation, all of which are influenced by glucose metabolism disruptions. Emerging evidence suggests that targeting glucose metabolism could offer therapeutic benefits for PD. Several antidiabetic drugs have shown promise in animal models and clinical trials for mitigating the symptoms and progression of PD. This review explores the current understanding of the association between PD and glucose metabolism, emphasizing the potential of antidiabetic medications as a novel therapeutic approach. By improving glucose uptake and utilization, enhancing mitochondrial function, and reducing neuroinflammation, these drugs could address key pathophysiological mechanisms in PD, offering hope for more effective management of this debilitating disease.

## 1. Introduction

Glucose metabolism is one of the most important biological processes in the body, essential for generating energy and maintaining cellular functions. Disruptions in glucose metabolism can lead to pathological conditions such as diabetes mellitus, emphasizing the importance of maintaining glucose homeostasis for overall health. Glucose metabolism is particularly critical for brain function, as the brain consumes about 20% of the body’s glucose-derived energy, despite comprising only about 2% of body weight [1]. Disruptions in glucose metabolism can lead to an insufficient energy supply for neurons, exacerbating neurodegenerative processes. Increasing evidence suggests that many neurodegenerative diseases, including Alzheimer’s disease (AD) and Parkinson’s disease (PD) are associated with abnormalities in glucose metabolism [2,3]. Understanding glucose metabolism in the brain is crucial not only for its role in energy production but also for its implications in various diseases. PD is characterized by the loss of dopaminergic (DA) neurons in the *substantia nigra* and α-synuclein (α-syn) protein aggregates forming Lewy bodies or Lewy neurites [4]. Although the pathogenesis of PD is not fully understood, it is known to involve several interrelated mechanisms. The loss of DA neurons leads to a dopamine deficiency in the striatum, disrupting the regulation of motor control. α-Syn protein aggregates contribute to neuronal dysfunction and death. Additionally, mitochondrial dysfunction, oxidative stress, neuroinflammation, and impaired protein degradation pathways are critical factors in PD progression [4]. Genetic mutations, such as those in the *SNCA*, *LRRK2*, *PARK2*, *PINK1*, *DJ-1*, and *GBA* genes, also contribute to familial forms of PD [5,6]. Emerging evidence suggests a significant relationship between glucose metabolism and PD [1,2,7,8]. In this review, we briefly introduce glucose metabolism in the central nervous system (CNS) and the impairment of glucose metabolism observed in PD. Additionally, we discuss several antidiabetic drugs that have been investigated in the treatment for PD, both in animal model systems and clinical trials.

## 2. Overview of Glucose Metabolism in the Brain

Glucose metabolism is essential for the maintenance and operation of the CNS [1]. This energy demand is due to the intensive brain activities, including maintaining ionic gradients, synaptic transmission, and supporting the overall functionality of neurons and glial cells. Glucose is transported into the brain via the blood–brain barrier (BBB) using glucose transporter proteins, primarily GLUT1, which is predominantly expressed on the endothelial cells of the BBB [9]. Once across the BBB, glucose is taken up by astrocytes and neurons through specific glucose transporters. Neurons primarily use GLUT3 for glucose uptake [10]. Given their high energy demands, neurons rely heavily on a continuous supply of glucose for their metabolic needs. GLUT3 has a higher affinity for glucose than GLUT1 [11]. This high affinity allows neurons to efficiently uptake glucose, even when glucose concentrations are low. Astrocytes utilize GLUT1 to absorb glucose from the extracellular space [12]. These glial cells play a crucial role in supporting neuronal function by metabolizing glucose through glycolysis. During CNS development, GLUT1 expression gradually increases to support brain growth and nutrient supply, while GLUT3 levels, which relate more closely to neuronal maturation and functional activity, also progressively rise and reach adult levels by postnatal days 21–30 in rat brains [13]. The transport of glucose into the brain depends on the concentration gradient between the blood and the brain extracellular fluid [14]. Under normal physiological conditions, the glucose concentration in the brain is lower than that in the peripheral blood, creating a gradient that drives glucose transport into the brain [15]. Once in the brain, glucose is rapidly utilized by neurons and glial cells for energy production, primarily through glycolysis and oxidative phosphorylation. Generally, there is a positive correlation between peripheral blood glucose levels and brain glucose levels [16]. When blood glucose levels increase, brain glucose levels also tend to rise due to the enhanced gradient for glucose transport. However, the relationship is not perfectly linear, as factors such as BBB permeability, the efficiency of glucose transporters, and regional brain activity can influence this correlation [17,18,19]. During intense neuronal activity, the brain may consume glucose more rapidly, affecting the local glucose concentration.

The brain has mechanisms to maintain glucose homeostasis. In cases of hypoglycemia, the brain prioritizes glucose uptake to protect neuronal function. Conversely, in hyperglycemia, the brain’s glucose uptake can become saturated, limiting further increases in brain glucose levels [20]. In individuals with diabetes, fluctuations in blood glucose levels can lead to corresponding changes in brain glucose levels, impacting cognitive function and potentially contributing to neurological complications. Acute hypoglycemia can significantly reduce brain glucose levels, leading to impaired cognitive function, seizures, and, in severe cases, neuronal damage [21]. Chronic hyperglycemia can lead to changes in the BBB and glucose transport mechanisms, potentially impacting brain health and increasing the risk of neurodegenerative diseases [22]. Astrocytes in the brain mainly carry out glycolysis, which is the process of converting glucose into lactate. This lactate is subsequently released to meet the energy requirements of neurons, especially during times of high demand or hypoxia [23]. Neurons take up this lactate through monocarboxylate transporters (MCTs), particularly MCT2, and convert it back into pyruvate, which enters the mitochondria for further energy production [24]. Neurons employ pyruvate from lactate or glycolysis to power mitochondrial ATP synthesis via oxidative phosphorylation, whereas astrocytes use the pentose phosphate pathway (PPP) to produce NADPH for antioxidant defense [25]. Under conditions of high metabolic demand, essential enzymes such as 6-phosphofructo-2-kinase/fructose-2,6-biphosphatase 3 (PFKFB3) can become compromised, limiting glycolysis in neurons and highlighting the need for astrocyte support [26]. Consequently, neurons predominantly rely on lactate from astrocytes for ATP production. The exact proportion of ATP derived from direct glycolysis versus astrocyte-supplied lactate varies depending on metabolic conditions and activity levels. During high neuronal activity, the reliance on astrocyte-derived lactate increases, highlighting the importance of the astrocyte–neuron lactate shuttle in supporting sustained neuronal function [27,28]. 

Under certain conditions, the brain can adapt its energy metabolism. During hypoglycemia or prolonged fasting, the brain can utilize alternative energy substrates like ketone bodies [26]. These substrates are transported across the BBB and metabolized in mitochondria to produce ATP, thereby supporting brain function when glucose availability is limited. Sodium-Glucose Cotransporter (SGLT) 1 and 2 also play roles in brain glucose metabolism, although their functions are not as extensively studied as in other tissues. SGLT1 is widely expressed in various brain regions, including the hippocampus, cerebral cortex, and hypothalamus. It is found on neurons and glial cells, facilitating active glucose transport into these cells against the concentration gradient [29]. This process is particularly crucial during conditions of low glucose availability, ensuring that neurons and glial cells receive sufficient glucose to meet their energy demands. SGLT1-mediated glucose uptake may support critical brain functions such as synaptic activity, plasticity, and neuroprotection, particularly during hypoglycemia or ischemic events. On the other hand, SGLT2 expression in the brain is less pronounced and primarily localized to the hypothalamus, amygdala, periaqueductal gray, and the nucleus of the solitary tract [30]. SGLT2 may play a role in regulating glucose levels in the cerebrospinal fluid, contributing to the overall metabolic environment necessary for neuronal function. Both SGLT1 and SGLT2 could have neuroprotective roles, especially under metabolic stress conditions, by aiding glucose transport and maintaining energy homeostasis. Glucose metabolism in the brain is summarized in Figure 1.

## 3. Regulatory Mechanisms of Glucose Metabolism

The brain’s energy demands are met through tightly regulated mechanisms that ensure a continuous supply of glucose and its efficient utilization. These regulatory mechanisms involve various levels of control, including hormonal regulation, cellular signaling pathways, and interactions between different cell types within the brain. Hormones play a crucial role in regulating glucose metabolism both systemically and within the brain. Insulin is critical in modulating glucose metabolism, cognitive function, and neuroprotection. Neurons and glial cells express insulin receptors [31,32], and insulin signaling enhances glucose uptake, promotes glycogen synthesis, and influences synaptic plasticity and neurotransmitter release. Glucagon, primarily regulating hepatic glucose production [33], indirectly affects brain glucose levels by increasing blood glucose, ensuring a steady supply to the brain during fasting or hypoglycemia. Glucocorticoids, such as cortisol, are vital in stress responses, increasing gluconeogenesis and reducing glucose uptake in peripheral tissues to ensure glucose availability for the brain [34]. Incretins, like GLP-1, enhance insulin secretion and inhibit glucagon release, thereby modulating blood glucose levels. GLP-1 receptors in the brain promote neuroprotection, reduce neuroinflammation, and support cognitive function [35]. Thyroid hormones, such as T3 and T4, regulate basal metabolic rate and influence glucose metabolism by enhancing glucose uptake, glycolysis, and oxidative metabolism in the brain [36]. Thyroid dysfunctions can lead to cognitive impairments and affect brain metabolism, highlighting their importance in CNS glucose regulation. Together, these hormones orchestrate the finely tuned regulation of glucose metabolism in the brain, crucial for maintaining neuronal function and overall brain health.

Several intracellular signaling pathways regulate glucose metabolism in brain cells. These include AMPK, which responds to low ATP levels by promoting glucose uptake, glycolysis, and mitochondrial biogenesis, and the PI3K/Akt pathway, which is activated by insulin to enhance glucose uptake, transporter translocation, and glycolysis [37]. Under hypoxic conditions, Hypoxia-Inducible Factor (HIF) is stabilized and activates the transcription of genes involved in glycolysis, increasing glucose uptake and anaerobic metabolism to maintain energy production [38]. 

Neurons and astrocytes cooperate metabolically, with astrocytes responding to neuronal activity by increasing glycolysis and lactate production, which is shuttled to neurons to support their energy needs during increased firing rates [39]. Glucose transporter expression and activity are tightly regulated, with GLUT1 on BBB endothelial cells upregulated to ensure continuous glucose supply, and neuronal GLUT3 increasing in response to heightened activity and energy demands, facilitating greater glucose uptake [14].

The brain employs various feedback mechanisms to maintain energy homeostasis. Negative feedback loops involve the inhibition of glucose metabolism pathways when energy levels are sufficient [40]. High ATP levels inhibit glycolytic enzymes, reducing glucose breakdown, while energy scarcity triggers positive feedback to enhance glucose uptake and utilization. Low ATP levels activate AMPK, which then promotes glycolysis and glucose transport [41]. During hypoglycemia, the brain adapts by upregulating glucose transporters and shifting to alternative energy sources like ketone bodies [42,43]. Chronic brain inflammation disrupts glucose metabolism by impairing insulin signaling and glucose transporter expression, leading to reduced glucose uptake and contributing to neuronal dysfunction and degeneration [3].

## 4. Abnormalities in Glucose Metabolism in PD 

DA neurons in the substantia nigra pars compacta (SNpc) are highly vulnerable to oxidative stress, which plays a critical role in PD progression. This vulnerability is due to their high metabolic demands necessitating significant ATP production via oxidative phosphorylation, a process that generates reactive oxygen species (ROS). Their unique electrophysiological properties, including frequent calcium influx, require substantial energy, further contributing to oxidative stress [44]. Consequently, DA neurons are more susceptible to abnormalities in glucose metabolism, further exacerbating their vulnerability and contributing to their degeneration. Moreover, many genes associated with familial PD, such as PARK2, PINK1, and DJ-1, are intimately linked to mitochondrial function [45]. These genes are vital for mitochondrial health, dynamics, and protection against oxidative stress, all of which are crucial for proper glucose metabolism through oxidative phosphorylation. Dysfunction in these processes can lead to impaired glucose utilization and ATP production, heightening the risk of glucose metabolism abnormalities in PD. This underscores the strong link between PD and disrupted glucose metabolism.

## 5. Abnormalities in Glucose Metabolism Observed in Human Studies 

Several studies have established a link between diabetes and an increased risk of developing PD. Retrospective and population-based cohort studies have independently associated prediabetes and elevated fasting glucose levels with an increased risk of PD [46,47]. Moreover, De Pablo-Fernandez and colleagues reported an increased incidence of PD in patients with type 2 diabetes compared to a reference cohort [48]. Prospective studies in Finland and the United States have further corroborated these findings, reporting that type 2 diabetes patients have a significantly increased risk of PD, by 85% and 40%, respectively [49,50]. A recent meta-analysis has further confirmed that diabetes can elevate the risk of PD [51]. 

Beyond increasing the risk of PD development, diabetes may also contribute to its progression. PD patients with diabetes typically present more severe motor symptoms and exhibit worse responsiveness to DA medications [52,53]. There is evidence to suggest that insulin resistance, a common feature of type 2 diabetes, may also occur in PD. Insulin resistance can impair glucose uptake and utilization in the brain, potentially exacerbating neurodegeneration. Studies have shown a marked loss of insulin receptor mRNA in the SNpc of patients with PD and increased insulin resistance compared with age-matched controls [54]. 

PD patients have been reported to exhibit altered glucose uptake in various regions of the brain. Fluorodeoxyglucose positron emission tomography (FDG PET) has revealed a PD-related pattern of metabolism characterized by relatively increased metabolism in certain brain regions (e.g., globus pallidus, putamen, thalamus, cerebellum, pons, and sensorimotor cortex) and relatively decreased metabolism in others (e.g., posterior temporoparietal occipital areas and sometimes the lateral frontal area, especially in PD with cognitive impairment) [55,56]. In addition, PD patients with GBA mutations exhibit further metabolic abnormalities, such as significant hypometabolism in the supplemental motor area and, in cases with parkinsonism, additional hypometabolism in the parietooccipital cortices [57]. 

Changes in glycolytic activity have also been observed in PD patients. Some studies have reported elevated lactate levels in the cerebrospinal fluid of PD patients [58]. Phosphoglycerate kinase-1 (PGK-1), a crucial enzyme in the glycolytic pathway, has been implicated in early-onset parkinsonism. PGK-1 deficiency, an X-linked recessive disorder, leads to various neurological disorders due to insufficient ATP regeneration, and early-onset parkinsonism has been occasionally reported as a complication of this condition [59]. Furthermore, reduced levels of glucose-6-phosphate dehydrogenase and 6-phosphogluconate dehydrogenase, key enzymes in the PPP, have been found in the putamen of early-stage PD and in the cerebellum of both early- and late-stage PD [60]. The contribution of gluconeogenesis to total glucose production is increased in idiopathic PD patients compared with healthy controls [61].

Studies have identified a deficiency in complex I of the mitochondrial electron transport chain in the substantia nigra of PD patients [62]. This deficiency leads to reduced ATP production and an increased production of ROS, contributing to oxidative stress and neuronal damage. The resulting impaired mitochondrial function and decreased ATP levels are insufficient to meet the high energy demands of DA neurons, potentially leading to neuronal dysfunction and cell death [63]. 

Chronic neuroinflammation, a prominent feature of PD [64], can significantly impact glucose metabolism. Inflammatory cytokines, such as TNF-α and IL-6, can interfere with insulin signaling pathways [65,66], promoting insulin resistance and impairing glucose uptake by neurons and glial cells. Activated microglia, the brain resident immune cells, can produce inflammatory mediators that disrupt glucose metabolism, further exacerbating neuronal injury and dysfunction.

## 6. Abnormalities in Glucose Metabolism in Model Systems

Chronic hyperglycemia has been demonstrated to induce DA neurodegeneration, leading to PD-like motor impairments in animal models [67,68]. In MPTP-treated mice, the localization and levels of GLUT1 have been reported to remain unaltered [69]. However, contrasting reports have shown decreased GLUT1 in the striatum of PD mouse models [70]. The degeneration of DA neurons is positively correlated with disruptions in metabolic connectivity in the MPTP plus probenecid mouse model, highlighting the impact of metabolic disturbances on the progression of PD [71]. The overexpression of hexokinase 2 has been found to protect against neurodegeneration in the rotenone and MPTP mouse models of PD by promoting glycolysis [72].

Chronic hyperglycemia exacerbates α-syn aggregation and DA neuronal loss in a PD model [73], suggesting that elevated glucose levels enhance oxidative stress and inflammatory responses, promoting α-syn aggregation and accelerating neuronal damage. α-Syn aggregates impair mitochondrial function by disrupting mitochondrial metabolism [74], leading to reduced mitochondrial efficiency and increased ROS production. This dysfunction is closely linked to disturbances in cellular glucose metabolism. α-Syn overexpression, especially its A53T mutant form, in SH-SY5Y human neuroblastoma cells induces amyloid aggregate formation and significantly impairs glycolysis. This occurs through reduced glyceraldehyde-3-phosphate dehydrogenase (GAPDH) activity and decreased glucose uptake and lactate production [75]. Conversely, wild-type human α-syn overexpression in transgenic mice causes significant metabolic abnormalities, including reduced body weight and adiposity, altered feeding behavior, and decreased energy expenditure [76]. Insulin resistance significantly reduces SNCA expression in insulin-resistant C2C12 myoblast and skeletal muscle tissues of type 2 diabetic mice. This suggests a negative correlation between insulin resistance and SNCA expression, highlighting the role of α-syn in enhancing glucose uptake via the PI3K/Akt signaling pathway [77]. These findings underscore the critical interplay between α-syn expression or aggregation and glucose metabolism, suggesting that metabolic dysregulation could be both a consequence and a contributing factor in the pathogenesis of PD, thereby offering potential therapeutic targets for managing the disease. 

PARK2 deficiency increases glycolysis and reduces mitochondrial respiration, driving the Warburg effect. Restoring PARK2 expression reverses these metabolic changes, indicating PARK2’s crucial role in p53-mediated energy metabolism and tumor suppression [78]. PARK2 knockout (KO) neurons also show significant disruptions in glucose metabolism, affecting glycolytic pathways and mitochondrial respiration [79]. A proteomic study on PINK1 KO rat brain documented changes in glycolysis-related proteins, with increased lactate dehydrogenase and reduced pyruvate kinase M levels [80]. PINK1 deficiency in β cells leads to increased basal insulin secretion and improved glucose tolerance in mice [81], and increased cell proliferation by reprogramming glucose metabolism through HIF1 [82]. PINK1 functions as a metabolic sensor linking glucose metabolism to mitochondrial quality control. In glucose-deficient or low-ATP conditions, PINK1 translation is inhibited, impairing mitophagy [83]. DJ-1 negatively regulates glycolysis and cell proliferation by transcriptionally upregulating PINK1 [84]. DJ-1 loss leads to an age-dependent accumulation of hexokinase 1 in the cytosol and activation of the polyol pathway of glucose metabolism in the brain [85]. LRRK2 influences glucose metabolism through its impact on insulin signaling and glucose transport mechanisms. It inhibits GLUT4 translocation to the plasma membrane by phosphorylating Rab GTPases, which are critical for insulin-dependent glucose uptake in adipocytes and muscle cells [86,87,88]. Abnormalities in glucose metabolism in PD are summarized in Table 1. 

## 7. Types of Major Antidiabetic Drugs and Their Mechanisms of Action and Effects of Antidiabetic Drugs in PD 

Antidiabetic drugs are primarily used to manage blood glucose levels in patients with diabetes. Given the growing interest in their potential neuroprotective effects, understanding their mechanisms of action is crucial. This section outlines the major types of antidiabetic drugs and their potential as novel therapeutic agents for PD. 

### 7.1. Insulin and Insulin Analogs

Insulin therapy is essential for patients with type 1 diabetes and is also used in advanced type 2 diabetes when other medications fail to maintain adequate glucose control. Although there is limited evidence for PD, a study by Novak and colleagues showed that intranasal short-acting (regular) insulin treatment improved motor performance and function compared to placebo, resulting in lower disability scores (Hoehn and Yahr scale) and improved Unified Parkinson’s Disease rating scale (UPDRS) motor scores compared to placebo [89].

### 7.2. Biguanides (Metformin) 

Metformin is the first-line treatment for type 2 diabetes due to its efficacy, safety profile, and additional benefits like weight loss and improved lipid profiles [90,91]. It activates AMPK, which improves peripheral glucose uptake, lowers intestinal glucose absorption, and increases insulin sensitivity [92,93]. AMPK activation also enhances mitochondrial biogenesis and function, reduces oxidative stress, and inhibits the mammalian target of rapamycin (mTOR) pathway [94], which may be involved in neuroprotection. Metformin can penetrate the BBB [95]. It has demonstrated protective effects against DA neuron loss and improved motor function through inhibiting neuroinflammation and ER stress in a rotenone-induced PD mouse model [96], improving autophagy in an MTPT-induced PD mouse model [97], and reducing mitochondrial respiration in a *C. elegans* model [98]. It has also been shown to reduce DA neuronal loss and α-syn phosphorylation, a hallmark of PD pathology in a subchronic MPTP model of PD [99]. In a neuroinflammation-based rat model of PD produced by LPS injection, metformin was observed to lower mRNA expression levels of TNF-α, IL-1, IL-6, iNOS, MCP-1, CD200, and CX3CR1 in the substantia nigra [100]. Clinically, numerous observational studies and retrospective analyses have suggested that the use of metformin is linked to a lower risk of mortality in diabetic patients with PD [101]. Research further revealed that metformin-treated diabetic patients had a reduced risk of PD compared to non-metformin-treated individuals [102]. These benefits are thought to be mediated by metformin’s effects on mitochondrial function, insulin sensitivity, and neuroinflammation [98,103]. Clinical trials are currently investigating metformin’s potential as a disease-modifying therapy for PD (NCT05781711). 

### 7.3. Sulfonylureas

Sulfonylureas are used in type 2 diabetes to enhance endogenous insulin secretion, especially in patients with residual beta-cell function [104]. Sulfonylureas stimulate insulin secretion from pancreatic beta cells by binding to and inhibiting the ATP-sensitive potassium channels (K_ATP channels) on the β cell membrane [105]. This inhibition leads to cell depolarization, calcium influx, and insulin release. Sulfonylureas stimulate insulin secretion, which may improve insulin signaling in the brain [106,107]. Improved insulin signaling can enhance neuronal survival and function via an alternatively spliced protein kinase CδII isoform [108]. Glimepiride prevents paraquat-induced PD pathology in mice, which is involved in oxidative stress and neuroinflammation [109]. Glimepiride ameliorates MPTP-induced PD motor and non-motor deficits through the enhancement of antioxidant defense signaling and attenuation of neuroinflammatory markers [110]. Abdelkader et al. also demonstrated that 3 mg/kg glibenclamide (i.p.) for 3 consecutive weeks exhibited evidently neuroprotective effects in rotenone-induced PD mouse model, possibly by anti-inflammatory and anti-apoptotic effects [111]. In A53T α-syn transgenic mice, SUR1, the K_ATP_-channel regulatory subunit that binds sulfonylurea, was upregulated, accompanied by neuronal damage. After interference with SUR1 expression by an injection of lentivirus into the substantia nigra, the progression of DA neuron degeneration was delayed [112]. There is also limited clinical evidence on the effects of sulfonylureas in PD patients. Genetic variants in the ABCC8 and KCNJ11 genes that increase the affinity of the subunits of the K_ATP_-channel to the effects of sulfonylureas are associated with a markedly lower risk of PD [113]. The mRNA levels of the SUR1 subunit were found to be approximately twofold higher in DA neurons from patients with PD than in those from individuals in the control group, as determined by quantitative mRNA expression profiling techniques [114]. Nevertheless, the limited ability of sulfonylureas to penetrate the blood–brain barrier may restrict their use in the treatment of PD [115]. More research is needed to explore the potential neuroprotective mechanisms of sulfonylureas in PD.

### 7.4. Thiazolidinediones (TZDs) 

TZDs including pioglitazone and rosiglitazone are used in type 2 diabetes to improve insulin sensitivity, but their use is limited by side effects such as weight gain, edema, and potential cardiovascular risks [116]. TZDs act as agonists for peroxisome proliferator-activated receptor-gamma (PPAR-γ) [117]. The activation of PPAR-γ enhances insulin sensitivity by increasing the storage of fatty acids in adipocytes, thereby reducing lipotoxicity in muscle and liver, and by improving glucose uptake [118]. Rosiglitazone has limited blood–brain barrier permeability (9–14%) [119,120], while pioglitazone can cross the blood–brain barrier [121]. Oral pioglitazone treatment significantly improved MPTP-damaged behavior symptoms, increased tyrosine hydroxylase (TH)-positive neuron survival, enhanced PGC-1*α* expression, and improved mitochondrial quantity and ultrastructure. In vitro, 2,4-thiazolidinedione increased mitochondrial function molecules (PGC-1*α*, NRF1, NRF2, and Mfn2), inhibited Fis1, and reversed MPP+-induced reductions in Bcl-2 and ERK, while Bax levels showed opposite changes [122]. In the MPTP-induced PD model, rosiglitazone fully prevented motor and olfactory dysfunctions, TH-positive cell loss, and microglia reactivity, while partially preserving striatal DA and inhibiting astroglia response [123,124]. Pioglitazone improves motor symptoms and protects against neurodegeneration in the MPTP-induced PD model, but not in the 6-OHDA rodent model [125]. Clinical trials with pioglitazone in PD patients have yielded mixed results. While some studies suggest potential benefits [126,127], others have not shown significant improvements [128,129]. The ADAGIO study investigated the effects of pioglitazone in PD patients but did not show significant benefits in slowing disease progression [128]. Individual cases have reported symptomatic improvements in PD patients taking pioglitazone. These observations align with preclinical findings that suggest PPAR-γ activation can reduce neuroinflammation and oxidative stress [124]. Ongoing research aims to clarify the role of thiazolidinediones in PD, with a focus on identifying patient subgroups that may benefit the most from these drugs [126]. 

### 7.5. Dipeptidyl Peptidase-4 (DPP-4) Inhibitors

DPP-4 inhibitors, such as sitagliptin and saxagliptin, are used in type 2 diabetes to improve glycemic control with a low risk of hypoglycemia [130,131]. DPP-4 inhibitors prevent the degradation of incretin hormones (GLP-1 and GIP), which enhance glucose-dependent insulin secretion and inhibit glucagon release [132,133,134], resulting in improved glucose control. DPP-4 inhibitors like sitagliptin and saxagliptin improve glucose control and reduce inflammation in type 2 diabetes, potentially offering neuroprotective benefits in PD through various molecular pathways. Vildagliptin mitigates MPTP-induced motor deficits and DA neuronal apoptosis by modulating PI3K/Akt, ERK, and JNK signaling pathways, and inhibiting autophagy in mouse and SH-SY5Y cell models [135]. Sitagliptin also alleviates DA neurodegeneration, neuroinflammation, and behavioral impairment in the rat 6-OHDA model of PD [136]. While preclinical data are promising, clinical evidence in PD patients is still limited, but some studies have suggested potential benefits [137]. A population-based cohort study by Brauer et al. indicated a lower incidence of PD in patients using DPP-4 inhibitors compared to those using other antidiabetic drugs [138]. Anecdotal reports have noted improvements in PD symptoms in patients treated with DPP-4 inhibitors for diabetes [138]. These findings suggest that enhancing incretin signaling may offer neuroprotective benefits [139].

### 7.6. Glucagon-like Peptide-1 (GLP-1) Receptor Agonists

GLP-1 receptor agonists, such as exenatide and liraglutide, are used in type 2 diabetes for their potent glucose-lowering effects and benefits on weight loss and cardiovascular health [140,141]. They enhance glucose-dependent insulin secretion, suppress glucagon secretion, slow gastric emptying, and promote satiety, leading to weight loss [142,143,144]. They can cross the blood–brain barrier in low dose form [145,146,147] and mimic the effects of the incretin hormone GLP-1. Additionally, GLP-1 receptor agonists demonstrate neuroprotective effects by activating NF-κB and PI3K/Akt pathways, reducing neuroinflammation through cytokine modulation, and inhibiting TNFα-induced monocyte adhesion while activating AMPK via CaMKKβ [148,149]. In PD models, the dual agonist DA-JC1 exhibits neuroprotective effects by reversing MPTP-induced motor impairments, preserving DA neurons, and increasing BDNF levels [150,151]. GLP-1 receptor agonists have been reported to reduce DA neuron loss, improve motor function, and decrease neuroinflammation and oxidative stress in preclinical studies [152,153,154]. Notably, a clinical study by Athauda et al. demonstrated significant motor function improvements in PD patients treated with exenatide compared to placebo [137]. Case reports have also documented improvements in PD symptoms and quality of life with exenatide treatment [155]. Exenatide has shown promise in clinical trials for improving motor symptoms and potentially slowing PD progression [156]. A recent phase 2 trial reported that lixisenatide reduced motor disability progression, albeit with some gastrointestinal side effects [157]. Ongoing trials aim to confirm these findings and assess the long-term effects of GLP-1 receptor agonists like liraglutide and semaglutide on PD progression (NCT02953665, NCT03659682, and NCT04269642). 

### 7.7. Sodium-Glucose Cotransporter-2 (SGLT-2) Inhibitors

SGLT-2 inhibitors, such as canagliflozin, dapagliflozin, and empagliflozin, are used in type 2 diabetes for their glucose-lowering effects, weight loss, and cardiovascular and renal benefits [158,159,160]. SGLT-2 inhibitors inhibit the SGLT-2 protein in the proximal tubules of the kidneys, reducing glucose reabsorption and increasing glucose excretion in the urine [161,162,163,164,165]. This leads to lower blood glucose levels and caloric loss. Most SGLT2 inhibitors are lipid-soluble and effectively cross the blood–brain barrier [166]. Empagliflozin prevented cognitive impairment in db/db mice by reducing cerebral oxidative stress via decreased NADPH oxidase subunits (gp91), increasing brain-derived neurotrophic factor, and ameliorating albuminuria and glomerular injury [167,168,169], which may provide neuroprotective benefits. Limited studies have investigated the effects of SGLT-2 inhibitors in PD models. However, empagliflozin mitigates neurodegeneration in rotenone-induced PD in rats by providing antioxidant and anti-inflammatory effects and modulating α-syn and PARK2 levels [170]. Empagliflozin counteracts neurodegeneration in rotenone-induced PD in rats by improving locomotor activity, reducing α-syn accumulation, reversing oxidative stress and inflammation, and activating the AMPK/SIRT-1/PGC-1α and Wnt/β-catenin pathways [171]. Dapagliflozin also attenuates neuronal injury and motor dysfunction in a rotenone-induced PD rat model via ROS-dependent AKT/GSK-3β/NF-κB and DJ-1/Nrf2 pathways [172]. Limited case reports have described improvements in metabolic parameters and overall well-being in PD patients treated with SGLT-2 inhibitors [173]. Further research is needed to explore the potential neuroprotective effects of SGLT-2 inhibitors in PD patients.

### 7.8. Alpha-Glucosidase Inhibitors

Alpha-glucosidase inhibitors, such as acarbose and miglitol, inhibit the alpha-glucosidase enzyme in the small intestine, delaying carbohydrate digestion and glucose absorption, thereby reducing postprandial blood glucose spikes [174,175]. These agents are used to manage postprandial hyperglycemia in type 2 diabetes [176]. Research on alpha-glucosidase inhibitors in PD is still limited. However, studies related to the effects of alpha-glucosidase inhibitors on general metabolism suggest that these drugs may have potential neuroprotective effects [177,178]. Since alpha-glucosidase inhibitors have a limited ability to cross the BBB, any neuroprotective effects they might provide would primarily be due to their impact on improving blood glucose levels [179]. 

## 8. Evaluating the Potential of Therapeutic Strategies Targeting Glucose Metabolism Against PD

Therapeutic strategies targeting glucose metabolism hold significant potential for mitigating the symptoms and progression of PD. By enhancing glucose uptake and utilization, improving mitochondrial function, reducing neuroinflammation, and utilizing alternative energy substrates, these approaches offer a multifaceted strategy to address the metabolic dysfunctions associated with PD. Moreover, recent advancements include the development of drugs that inhibit key enzymes in glucose metabolism, such as lactate dehydrogenase and pyruvate dehydrogenase, to counteract neurodegeneration [180,181]. These enzyme inhibitors represent promising candidates for future PD therapies and may broaden the scope of metabolic interventions for neurodegenerative conditions. Continued research and clinical trials are essential to validate these strategies and optimize their therapeutic potential. The effects of antidiabetic drugs in PD are summarized in Table 2.

## 9. Conclusions

Glucose metabolism is essential for maintaining neuronal function and energy balance in the brain. Disruptions in glucose metabolism are increasingly recognized as significant contributors to the pathogenesis of neurodegenerative diseases such as PD. The connection between glucose metabolism and PD is supported by clinical and preclinical studies, which demonstrate that alterations in insulin signaling, mitochondrial dysfunction, oxidative stress, and neuroinflammation significantly affect disease progression. However, there are varying results among studies, highlighting the complexity of glucose metabolism in different cellular contexts. While rodent models used in preclinical studies have provided substantial insights into PD pathogenesis, there is a recognized gap between rodent models and clinical presentations. To address this, non-human primate models are being utilized to bridge this gap as they replicate key characteristics of human PD, including α-syn accumulation and age-associated neurodegeneration [182]. The use of non-human primate models can thus enhance the translational relevance of preclinical findings and validate novel therapeutic strategies. Additionally, recent advancements in single-cell RNA sequencing (scRNA-seq) have provided new opportunities to explore cell-specific molecular mechanisms involved in PD pathogenesis [183]. While direct findings on glucose metabolism dysfunction using scRNA-seq are limited, this technology holds promise for future investigations into metabolic pathways at a single-cell level, potentially uncovering new therapeutic targets. 

The repurposing and development of antidiabetic drugs that target glucose metabolism offers a promising therapeutic approach for PD. Drugs such as metformin, sulfonylureas, and thiazolidinediones have shown potential in preclinical PD models and are currently under investigation in clinical trials. Further research is needed to clarify their mechanisms of action in the brain, optimize dosing regimens, and identify patient subgroups most likely to benefit from these therapies. As our understanding of glucose metabolism and its role in neurodegeneration continues to evolve, targeted interventions to modulate glucose homeostasis may offer new hope for delaying or halting PD progression. 

## Figures and Tables

**Figure 1 cells-13-01876-f001:**
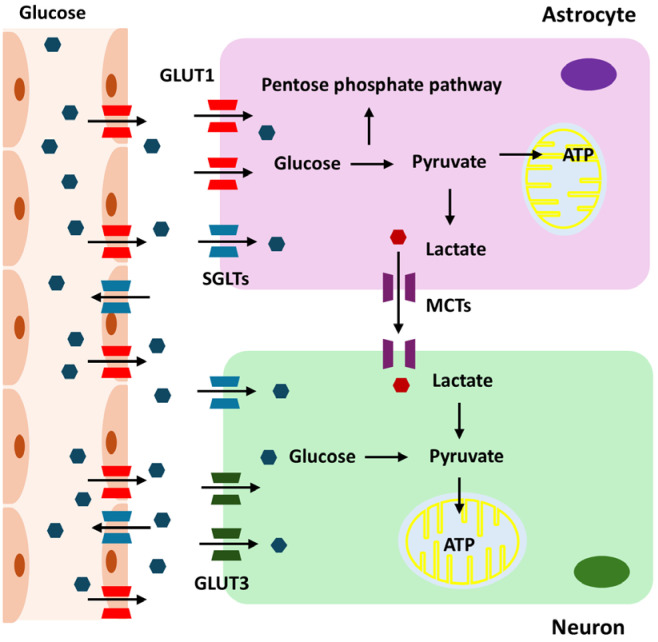
Overview of glucose metabolism in the brain.

**Table 1 cells-13-01876-t001:** Glucose metabolism abnormalities in PD.

Observed Abnormalities	Details
Human Studies	
Diabetes and PD Risk	Association between elevated fasting glucose and PD risk [46,47] Increased incidence of PD in patients with type 2 diabetes [48] Increased risk of PD in diabetic patients [49,50,51]
Diabetes Impact on PD Progression	More severe motor symptoms in PD patients with diabetes [52,53] Worse responsiveness to DA medications [52]
Insulin Resistance	Decreased insulin receptor mRNA in substantia nigra pars compacta of PD patients [54] Increased insulin resistance compared to age-matched controls [54]
Brain Glucose Metabolism Changes	FDG PET shows pattern of increased and decreased metabolism in specific brain regions [55,56] Elevated lactate levels in cerebrospinal fluid [58] Early-onset parkinsonism in PGK-1 deficiency [59] Reduced levels of glucose-6-phosphate dehydrogenase and 6-phosphogluconate dehydrogenase in the putamen of early-stage PD and in the cerebellum of both early- and late-stage PD [60]
Mitochondrial Dysfunction	Deficiency in complex I of the electron transport chain [62] Reduced ATP production and increased ROS generation [62]
Model systems	
Effects of Chronic Hyperglycemia	Induces DA neurodegeneration in a rat or 6-OHDA mouse model [67,68] Leads to PD-like motor impairments in a rat or 6-OHDA mouse model [67,68]
Toxin Models	Disruptions in metabolic connectivity in the MPTP plus probenecid mouse model [71] Overexpression of hexokinase 2 protects against neurodegeneration in the rotenone and MPTP mouse models [72]
α-Syn Related Changes	Hyperglycemia exacerbates α-syn aggregation and neuronal loss [73] α-Syn aggregates impair mitochondrial function by disrupting mitochondrial metabolism [74] α-Syn overexpression impairs glycolysis [75] Insulin resistance significantly reduces SNCA expression in insulin-resistant C2C12 muscle cells and skeletal muscle tissues of type 2 diabetic mice [77]
PD Gene-Related Changes	PARK2 deficiency increases glycolysis and reduces mitochondrial respiration [78,79] PINK1 deficiency changes in glycolysis-related proteins [80] PINK1 deficiency in β cells leads to increased basal insulin secretion and improved glucose tolerance in mice [81] PINK1 functions as a metabolic sensor, linking glucose metabolism to mitochondrial quality control [83] DJ-1 negatively regulates glycolysis [84] DJ-1 loss leads to the accumulation of hexokinase 1, and activation of polyol pathway [85] LRRK2 influences glucose metabolism [86,87,88]

**Table 2 cells-13-01876-t002:** Effects of antidiabetic drugs in PD.

Drug Type	Mechanism of Action	Effects in PD Models	Clinical Evidence in PD
Insulin and Insulin Analogs	Activates the PI3K/Akt pathway to promote glucose uptake, glycogen synthesis, and lipid synthesis, and to inhibit glucose production	Limited evidence in PD models	Improved motor performance and function [89]
Biguanides (Metformin)	Activates AMPK and improves insulin sensitivity	Protects against DA neuron loss [96] Improves motor function [96] Improves autophagy [97] Reduces mitochondrial respiration to control levels [98] Reduces α-syn phosphorylation [99] Lowers inflammatory markers [100]	Lower risk of PD in diabetic patients [102] Lower mortality risk in PD patients with diabetes [101]
Sulfonylureas	Enhances insulin secretion	Prevents paraquat-induced PD pathology [109] Ameliorates MPTP-induced motor and non-motor deficits [110] Neuroprotective effects in rotenone-induced PD [111] Delays DA neuron degeneration in A53T α-syn transgenic mice [112]	Genetic variants in the ABCC8 and KCNJ11 genes associated with lower PD risk [113] Higher SUR1 mRNA levels in PD patients [114]
Thiazolidinediones (TZDs)	Improves insulin sensitivity via PPAR-γ activation	Improves MPTP-damaged behavior [122] Increases TH-positive neuron survival [122] Enhances mitochondrial function [122] Prevents motor and olfactory dysfunctions [123,124] Reduces microglia reactivity [123,124]	Mixed results in clinical trials [128] Some case reports of symptomatic improvements [126]
DPP-4 Inhibitors	Prevents degradation of incretin hormones	Mitigates MPTP-induced motor deficits [135] Alleviates DA neurodegeneration [136] Reduces neuroinflammation [136]	Lower incidence of PD in users [138] Anecdotal reports of symptom improvements [138]
GLP-1 Receptor Agonists	Enhances glucose-dependent insulin secretion and suppresses glucagon	Reduces DA neuron loss [152,153] Improves motor function [152,154] Decreases neuroinflammation [152] Increases BDNF levels [150]	Significant motor function improvements in clinical trials [137] Improvements in PD symptoms and quality of life [155] Potential to slow disease progression [156]
SGLT-2 Inhibitors	Reduces glucose reabsorption in kidneys	Mitigates neurodegeneration in rotenone-induced PD [170] Improves locomotor activity [171] Reduces α-syn accumulation [171] Activates AMPK/SIRT-1/PGC-1α and Wnt/β-catenin pathways [171] Attenuates neuronal injury via ROS-dependent pathways [172]	Some observational studies suggest potential benefits including neuroprotective effects [173]
Alpha-Glucosidase Inhibitors	Delays carbohydrate digestion and glucose absorption	Limited research in PD models	No significant clinical evidence in PD

## Data Availability

Not applicable.

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
