# Peer review of "Targeting Glucose Metabolism: A Novel Therapeutic Approach for Parkinson’s Disease"

_cells, 2024, doi:10.3390/cells13221876_

Round 1
Reviewer 1 Report
Comments and Suggestions for Authors
Tanvir and coworkers summarized the association between Parkinson's Disease (PD) and glucose metabolism with emphasize on the potential of the current antidiabetic medications for potential therapeutics of PD. This paper was well-written and covered all essential aspects of the known connection between PD and glucose metabolism while maintaining concise in the phrasing. Thus I will recommend the publication of the present form.
Minor comment:
The Table II. Effects of Antidiabetic Drugs in PD is recommended to be reorganized for the readability. It is not comfortable to read a single word written across several lines (e.g. Thia-
zoli-
dinedi-
ones)
1. For the therapeutics the authors include in the manuscript, they are exclusively antidiabetic drugs. How about other therapeutics targeting the downstream pathways of glucose metabolism: such as lactate dehydratenase, pyruvate dehydrogenase kinase and fatty acid synthesis? The authors are recommended to also mention some of them. 2. I would also suggest the authors draw a glucose metabolism pathway graph in the neuronal systems and mark the targets of the selected therapeutics. 3. There is a recent clinical trial of lixisenatide (GLP1-receptor agonist) for treating early PD patients (https://www.nejm.org/doi/10.1056/NEJMoa2312323). It helps to add this in the reference list.
Author Response
Minor comment:
The Table II. Effects of Antidiabetic Drugs in PD is recommended to be reorganized for the readability. It is not comfortable to read a single word written across several lines (e.g. Thia-
zoli-dinedi-ones)
[Response] Thank you for your valuable feedback. As suggested by Reviewer #1, we have reorganized Table II to improve readability.
- For the therapeutics the authors include in the manuscript, they are exclusively antidiabetic drugs. How about other therapeutics targeting the downstream pathways of glucose metabolism: such as lactate dehydratenase, pyruvate dehydrogenase kinase and fatty acid synthesis? The authors are recommended to also mention some of them.
[Response] Thank you for your valuable feedback. As suggested by Reviewer #2, we have added these findings (see p11, lines 468-472).
- I would also suggest the authors draw a glucose metabolism pathway graph in the neuronal systems and mark the targets of the selected therapeutics.
[Response] Thank you for your valuable feedback. The primary purpose of Figure 1 was to provide a simplified overview that highlights the main cells and factors involved in glucose metabolism within the brain, ensuring clarity and a straightforward visual representation. We have intentionally kept it minimalistic to focus on the core components. Instead, to address your suggestion, we have modified the graphic abstract to include the targets of the antidiabetic drugs. While this addition has made the figure slightly more complex, we believe it provides a more comprehensive representation of the interactions within the glucose metabolism pathway in neuronal systems. We hope this adjustment enhances the visual and informational quality of the manuscript.
- There is a recent clinical trial of lixisenatide (GLP1-receptor agonist) for treating early PD patients (https://www.nejm.org/doi/10.1056/NEJMoa2312323). It helps to add this in the reference list.
[Response] Thank you for your valuable feedback. As suggested by Reviewer #1, we have added the recent clinical trial on lixisenatide (GLP1-receptor agonist) for treating early PD patients to the reference list (see p10, lines 425-426). This inclusion strengthens the discussion on current clinical advancements and ensures the manuscript reflects the latest developments in the field. We appreciate your suggestion and believe it enhances the comprehensiveness of our work.

Reviewer 2 Report
Comments and Suggestions for Authors
Dear authors,
Thank you for sharing the manuscript entitled “Targeting Glucose Metabolism: A Novel Therapeutic Approach for Parkinson's Disease”.
So far, there is no therapy can halt or reverse PD pathology progression, as well as prevention. Emerging evidences indicated that diabetes can increase PD incidence, hinting glucose as culprit for PD pathogenesis. Brain consumes around 20% of human body energy, primarily supplied by glucose, while only amounts for 2% body weight. Most PD related gene mutations are energy relevant. Clinical trials suggest anti-diabetes drugs have potential for PD benefit. These facts convergently imply gluose is a potential traget for PD treatment. Your manuscript well summarized recent findings in this field, it is good for students or investigators who are entering this field.
I would like to suggest some minor adjustments for current manuscript:
1. Drug induced animal model, such as MPTP and rotenone, are widely used model for PD research, I think this information should be integrated into table I;
2. Although rodent models contribute enormously to understand PD pathogenesis, there is a conspicuous gap between clinic and rodent models. Thus, primate PD models should be mentioned, at least in conclusion section;
3. As single cell sequencing technology advanced, offering researchers new tool to uncover PD pathogenesis at cell-specific molecular level, also is chance to confirm glucose related hypothesis or investigate glucose contribution. So I would recommend to add some findings in PD patients or animal model discovered by single cell sequencing concerning glucose metabolism. If there is no report on it, talk potential contribution in conclusion section.
4. Offer more insightful views on potential molecular mechanism in conclusion section, especially for those controversial results from different studies.
Author Response
Response to Reviewer #2
I would like to suggest some minor adjustments for current manuscript:
1. Drug induced animal model, such as MPTP and rotenone, are widely used model for PD research, I think this information should be integrated into table I;
[Response] Thank you for your valuable feedback. As suggested by Reviewer #2, we have integrated information on drug-induced animal models, such as MPTP and rotenone, into Table I. These toxin models, which we have described in detail in the main text, are widely recognized for their utility in PD research and provide important insights into disease mechanisms and potential therapeutic approaches. We appreciate your suggestion to enhance the comprehensiveness of our table.
2. Although rodent models contribute enormously to understand PD pathogenesis, there is a conspicuous gap between clinic and rodent models. Thus, primate PD models should be mentioned, at least in conclusion section;
[Response] Thank you for your valuable suggestion. In response, we have expanded the conclusion section to include a discussion on the importance of non-human primate (NHP) models in PD research (p12, lines 485-490). We believe that this addition strengthens the conclusion and highlights the importance of using NHP models to bridge preclinical and clinical research.
- As single cell sequencing technology advanced, offering researchers new tool to uncover PD pathogenesis at cell-specific molecular level, also is chance to confirm glucose related hypothesis or investigate glucose contribution. So I would recommend to add some findings in PD patients or animal model discovered by single cell sequencing concerning glucose metabolism. If there is no report on it, talk potential contribution in conclusion section.
[Response] We appreciate and agree with the reviewer’s insightful comments. Single-cell sequencing technology holds great promise for uncovering cell-specific mechanisms in PD, including glucose metabolism's role. While current studies employing this technology are increasing, our comprehensive review found no reports specifically addressing glucose metabolism dysfunction at the single-cell level in PD. We will add a statement in the conclusion section acknowledging this research gap and highlighting the importance of future studies to explore this area (p12, lines 491-496). This addition will emphasize the potential of such investigations and guide future research directions.
4. Offer more insightful views on potential molecular mechanism in conclusion section, especially for those controversial results from different studies.
[Response] Thank you for your valuable suggestion. We have revised the conclusion to provide more insights into potential molecular mechanisms and address controversial findings. We emphasized the importance of human primate models and single-cell RNA sequencing (scRNA-seq) for understanding cell-specific aspects of PD, which can help reconcile conflicting results. Additionally, we acknowledged discrepancies in studies on glucose metabolism and proposed using advanced methods like scRNA-seq to resolve them. We believe these revisions highlight the complexity of PD and suggest directions for future research. (see p12, conclusion section)

Reviewer 3 Report
Comments and Suggestions for Authors
The review is overall well-written.
Please find below my comments and suggestions.
Line 49. “Genetic mutations, such as those in the
SNCA, LRRK2, PARK2, PINK1, and DJ-1 genes, also contribute to familial forms of PD” - Names of genes should be italicized.
Speaking about genetics of PD, what about PD associated with pathogenic sequence variants in the GBA1 gene?
(please refer to https://www.mdpi.com/1422-0067/25/13/7102 or any other publications on this subject)
Re: Figure 1. In my opinion the figure is too simplistic. Only GLUTs are shown in this picture.
What about changes in GLUT levels in astrocytes and neurons during CNS development (if such data exist)?
What about cross-talk between astrocytes and neurons (as related to glucose metabolism) in PD-related Peripheral neuropathy? The authors tend to focus only on glucose metabolism in CNS.
Can transcriptome studies shed some light on the role of glucose metabolism (and pathways involved) in PD?
Author Response
Response to Reviewer #3
The review is overall well-written.
Please find below my comments and suggestions.
Line 49. “Genetic mutations, such as those in the
SNCA, LRRK2, PARK2, PINK1, and DJ-1 genes, also contribute to familial forms of PD” - Names of genes should be italicized.
[Response] Thank you for your valuable feedback. As suggested by Reviewer #3, we have corrected them (see p2, line 53).
Speaking about genetics of PD, what about PD associated with pathogenic sequence variants in the GBA1 gene?
(please refer to https://www.mdpi.com/1422-0067/25/13/7102 or any other publications on this subject)
[Response] Thank you for your question regarding the genetic aspects of PD, specifically related to pathogenic sequence variants in the GBA1 gene. We have incorporated a discussion on GBA1-associated PD, highlighting its significance as one of the most common genetic risk factors for developing PD (see p2, line 53, p5, lines 221-224).
Re: Figure 1. In my opinion the figure is too simplistic. Only GLUTs are shown in this picture.
What about changes in GLUT levels in astrocytes and neurons during CNS development (if such data exist)?
[Response] Thank you for your valuable feedback. The primary purpose of Figure 1 was to provide a simplified overview that highlights the main cells and factors involved in glucose metabolism within the brain, ensuring clarity and a straightforward visual representation. We have intentionally kept it minimalistic to focus on the core components. Regarding your suggestion about changes in GLUT levels in astrocytes and neurons during CNS development, we have incorporated relevant information into the main text to provide a more comprehensive discussion (p2, lines 72-75). This addition highlights how GLUT1 and GLUT3 expression levels evolve during development. We hope this update sufficiently addresses your concern and enhances the depth of the manuscript.
What about cross-talk between astrocytes and neurons (as related to glucose metabolism) in PD-related Peripheral neuropathy? The authors tend to focus only on glucose metabolism in CNS.
[Response] Thank you for your insightful comment and for highlighting the importance of investigating non-motor symptoms, such as peripheral neuropathy, in PD. We agree that non-motor symptoms play a crucial role in understanding the pathophysiology of PD and significantly impact the quality of life of patients. The cross-talk between astrocytes and neurons, particularly as it pertains to glucose metabolism, is indeed an essential aspect that could extend beyond the central nervous system (CNS) to involve peripheral systems. However, as you have pointed out, current research on the involvement of glucose metabolism in PD-related peripheral neuropathy is extremely limited. This scarcity of data presents challenges in addressing this topic comprehensively within the scope of this review. Furthermore, while peripheral neuropathy is an important area of research in the context of non-motor symptoms, including a detailed discussion on it would extend beyond the primary focus of this review, which centers on CNS mechanisms. We hope you understand the rationale for not including this aspect in the current manuscript and appreciate your suggestion as a valuable direction for future research that could provide more holistic insights into PD pathophysiology.
Can transcriptome studies shed some light on the role of glucose metabolism (and pathways involved) in PD?
[Response] As noted in our response to Reviewer #2, transcriptome studies, especially single-cell RNA sequencing (scRNA-seq), offer valuable insights into glucose metabolism and related pathways in PD. Although research using scRNA-seq is expanding, our review found no specific studies on glucose metabolism dysfunction at the single-cell level in PD. We will include a statement in the conclusion to acknowledge this gap and emphasize the need for future research (p12, lines 491-496).
